# Family Communication as a Mediator between Family Resilience and Family Functioning under the Quarantine and COVID-19 Pandemic in Arabic Countries

**DOI:** 10.3390/children10111742

**Published:** 2023-10-27

**Authors:** Aiche Sabah, Musheer A. Aljaberi, Jamel Hajji, Chuan-Yin Fang, Yu-Chen Lai, Chung-Ying Lin

**Affiliations:** 1Faculty of Human and Social Sciences, Hassiba Benbouali University of Chlef, Chlef 02076, Algeria; 2Faculty of Medicine and Health Sciences, Taiz University, Taiz 6803, Yemen; musheer.jaberi@gmail.com; 3Department of Community Health, Faculty of Medicine & Health Sciences, Universiti Putra Malaysia, Serdang 43300, Malaysia; 4Faculty of Nursing and Applied Sciences, Lincoln University College, Petaling Jaya 47301, Malaysia; 5Higher Institute of Sport and Physical Education of Gafsa, Gafsa University, Gafsa 2100, Tunisia; jamel5hajji@gmail.com; 6Faculty of Human and Social Sciences of Tunis, University of Tunis, Tunis 1007, Tunisia; 7Division of Colon and Rectal Surgery, Ditmanson Medical Foundation Chia-Yi Christian Hospital, Chiayi 621, Taiwan; 13335@cych.org.tw; 8Institute of Allied Health Sciences, College of Medicine, National Cheng Kung University, Tainan 701, Taiwan; cylin36933@gmail.com; 9Faculty of Social Sciences, Media and Communication, University of Religions and Denominations, Qom 37491-13357, Iran

**Keywords:** children, adolescents, family, family resilience, family communication, family functioning, psychosocial impacts, mental health, well-being, quarantine, COVID-19 pandemic

## Abstract

The coronavirus pandemic has become an unprecedented world crisis in which we have struggled against the most potent threat of the twenty-first century. This pandemic has had a profound impact on individuals and families. Therefore, the study aimed to examine family communication as a mediator of the relationship between family resilience and family functioning under the quarantine and coronavirus pandemic in Algeria and Iraq. This study was conducted among individuals in Iraq and Algeria (N = 361). The respondents completed the Family Communication Scale (FCS), Walsh Family Resilience Questionnaire (WFRQ), and Family Functioning Scale (FFS). Structural equation modeling (SEM) with the bootstrapping method was used to conduct the mediated effects of family communication. Using the bootstrapping method in SEM, family resilience and communication significantly affected family functioning (coefficient = 0.808). Moreover, the direct effect and indirect effect (via family functioning) of family resilience on family functioning were both significant, with coefficients of 0.682 and 0.126. In addition, numerous groups from Iraq and Algeria have been analyzed as a sample and have shown no differences in the relationships between family resilience, family communication, and family functioning. In conclusion, the results showed that family communication mediated the relationship between family resilience and family functioning. Moreover, the type of this mediation seemed to be partial because of the significant direct relationship between family resilience and family functioning. According to the findings, healthcare providers should consider improving family resilience and communication to achieve good family functioning.

## 1. Introduction

A pandemic is an infectious disease that spreads worldwide, causing sickness and death. This often leads to economic, social, and political crises as people travel and move around [1,2,3,4,5,6,7,8]. The COVID-19 pandemic is an infectious disease that has affected millions worldwide [9,10,11,12,13], with an increased risk of infection, disease-related complications, and mortality [14,15,16]. Despite vaccination efforts, the future remains uncertain [11,17,18,19], highlighting the potential for future pandemics even as this one ends [20,21,22]. Arab countries have also been impacted, with many confirmed cases [3,13,17,23,24,25,26,27], leading to implementing precautions like curfews and social distancing measures [28,29].

The effects of COVID-19 on mental health, such as increased anxiety and disruptions [30,31], have been significant. It has also impacted family and social life. Previous evidence indicates that the COVID-19 pandemic seriously threatens families’ mental health and well-being [25,29,32,33,34], especially adults and older people [35,36,37]. It has led to increased levels of anxiety and depression during and after the pandemic, with potential long-term effects [38,39,40,41]. Feelings of loneliness, worry, fear, stress, anxiety, and pressure have also increased [42,43,44]. Therefore, family support, positive coping strategies, communication, and social support are essential for better outcomes [43,45].

The consequences of difficulties in the family system are likely to have long-term effects because contextual risks are deeply embedded in the structures and processes of family interactions. Therefore, there is a need for further research to measure the impact of the epidemic on family well-being, as the interplay between family dynamics and COVID-19 procedures is complex. What may seem like simple strategies can unexpectedly present challenges, and how individuals are affected by these procedures depends on their family composition and societal context [29,46].

The pandemic’s preventive measures and movement restrictions have posed significant challenges and effects to the essential constructs of family communication, relationships, resilience, and functioning [47], resulting in negative emotions and lifestyle changes [20,48,49]. Among these challenges, family communication stands out as a vital issue. The absence of family members in hospitals due to COVID-19 has made family gatherings and critical decision-making complex [50]. Lockdowns and stay-at-home orders have disrupted daily life, increasing household pressure, household shifts, changes in family relationships, and communication issues [51,52]. Due to social distancing and lockdown policies, virtual communication has become common [53]. Studies indicate various risk factors and protective behaviors have influenced family resilience during the COVID-19 pandemic. It was found that families with risk factors had lower levels of resilience during the pandemic [54], and resilience was associated with relaxation strategies, household chore participation, confidence in coping, changes in sleep patterns, and personal attribution [55]. Family harshness and economic status also influenced resilience during the pandemic [56]. Previous studies indicate that the COVID-19 pandemic significantly affected family functioning and dynamics [57]. It was found that approximately 19% of participants reported a functional impairment in their families ranging from moderate to severe [58]. Restrictive measures had a negative impact on these family relationships [59,60], which rely on the strength of family cohesion [61]. Family cohesion during the pandemic was associated with the individual’s level of anxiety [62], with increased tension, changes in family dynamics, and shifts in caregiving responsibilities being noted [51].

From the literature reviewed in the context of family dynamics during the pandemic, the challenge lies in examining the links between family communication, resilience, and functioning and how the quality of intra-family communication affects resilience and family functioning. To study family dynamics, the Olson Circumplex Model provides a solid foundation for exploring the relationship between family resilience, communication, and functioning. This model has been studied across different family cultures and found to be applicable in various regions, including the United States, European countries, and Hong Kong. Emphasis has been placed on the importance of this model in studying family-related variables [63,64,65,66,67].

The pandemic has significantly impacted families, with various short- and long-term effects on family relationships, communication, resilience, and functioning. This research aims to provide a more comprehensive investigation into how the pandemic influences families. It highlights the importance of studying family functioning, including family communication, and its direct and indirect effects on family resilience and functioning within different segments of Arab societies. Specifically, this study will focus on Algeria and Iraq, which implemented quarantine measures during the pandemic.

### 1.1. The Direct Effect on Family Communication, Family Functioning, and Family Resilience

The quarantine and the pandemic have affected families in terms of communication and satisfaction, which in turn influence family functioning. Three variables are considered to evaluate healthy family processes: family communication, family functioning, and family resilience. Several decades of research have identified two strategies for maintaining a healthy family. The first strategy is result-oriented, focusing on a family’s functioning, while the second strategy is process-oriented, describing family functions [68]. One of the most prominent result-oriented theories is Olson’s Circumplex Model of Marital and Family Systems [63,69,70]. This model emphasizes three primary aspects within family systems: resilience, cohesion, and communication. The model suggests that well-balanced couples and families exhibit greater efficiency than their imbalanced counterparts [71]. It is widely accepted as a definition of a healthy family [72]. The proposed study model is grounded in Olson’s conceptual model of families, which considers communication as a mediating and “facilitating” variable between resilience and cohesion [72].

Family resilience refers to a family’s ability to deal with challenges and adversity. It involves returning from difficult situations with increased strength and resourcefulness, marked by endurance, self-correction, and growth in response to crises. Resilient families can adapt and thrive under stress, influenced by various factors, while maintaining their integrity and well-being. This involves rising above losses, staying flexible, and moving forward, recognizing that all families possess strengths and resources [73,74].

During a family crisis, its members’ behavior often changes and many may struggle to confront it. In such times, when family members need to support each other, the family unit itself is typically affected, making it challenging to return to its previous state. According to Walsh [75], the COVID-19 pandemic represents a significant source of stress for families, as it brings about feelings of loss due to the death of loved ones, a lack of social and physical contact with significant others, disruptions in life plans and rituals, and financial instability. These factors contribute to the expectation of long-term psychological reactions [76].

The COVID-19 crisis serves as a crucial period to assess family resilience and further strengthen it. Nevertheless, resilient families aim to confront adversity and emerge even stronger. A study by Sabah et al. [77] found a high level of family resilience. Notably, there are no significant variations in the fundamental aspects of family resilience between respondents in Iraq and Algeria. The only distinction is that the Iraqi group scored higher in three processes: making sense of adversity, providing clear and consistent messages, and engaging in collaborative problem-solving.

As an independent variable, family resilience plays a significant role in shaping family functioning. It provides a framework for understanding how families can confront and adapt to challenging situations, positively impacting their functioning. Family resilience can guide families through difficulties, nurturing their strengths and maintaining a positive outlook [78]. It also aids families in addressing challenges and adapting to stressful circumstances, ultimately enhancing overall family functioning [79]. Research shows that family resilience is associated with a family’s ability to face adversity [80]. Furthermore, it can help families confront and overcome challenges, reinforcing family bonds [73].

Family functioning refers to the social and practical aspects of familial connections that contribute to the overall well-being of its members. It includes various elements such as adaptation, responsiveness, cohesion, problem-solving, and development, all of which are crucial for assessing the potential success of the family unit. Financial provision, family structure, socialization, safeguarding vulnerable family members, role fulfillment, emotional receptiveness, involvement, behavioral regulation, harmony, mutual support, and conflict resolution are also part of family functioning. The unique attributes of parent–child relationships, such as parental nurturing and authority, also indicate a family’s functionality [81,82].

Moreover, family functioning involves the family members’ ability to collaborate and assist each other in achieving the family’s objectives and goals [83]. In Algeria, the study of family functioning encompasses four dimensions: coherence, interaction, commitment, and intra-family coping [84].

Effective communication is the cornerstone of family life, and functioning is crucial in shaping individuals [85]. It serves as a tool for parents and children to redefine their roles, nurture their relationships, and develop greater mutual understanding. Socialization within the family helps individuals acquire cultural and social norms essential to integrate into society [86,87,88]. Family communication involves sharing information, ideas, thoughts, and emotions among family members. It facilitates adaptation, cohesion, and resilience in families to meet development challenges and changing situations [72].

Communication is the third dimension in the Circumplex Model, which is crucial to understanding family dynamics. Effective communication among family members is how people express their needs and desires for one another. In our research, family communication is considered a mediating variable, assessed by examining family interactions such as listening, speaking, self-disclosure, clarity, continuity tracking, respect, and listening skills [63,70]. Effective communication is essential for strong and healthy families, while weak communication tends to be characteristic of less healthy family relationships [81]. Studies have shown that communication styles have a robust connection with the stability of family relations, and positive communication skills used within the family system can help families adapt their levels of family functioning [67,89,90,91,92].

According to the Circumplex Model [93], communication refers to the positive communication skills used within the family system. In contrast, the communication dimension is viewed as a stabilizing factor that helps families adapt their levels of family functioning. Similarly, Carr [94] suggested that family functioning, as assessed through Olson’s model, was the most reliable predictor of family resilience. Therefore, members of families characterized by high levels of resilience exhibit a balance between cohesion, open communication, and a general sense of satisfaction with the family.

### 1.2. Conceptual Framework and Research Question

The COVID-19 pandemic has caused significant disruptions in the lives of many families, leading to challenges and changes in family dynamics [95,96]. Due to lockdown measures, family members have been confined together, increasing tension and conflicts [97]. The pandemic has also impacted family relationships and resulted in a reassignment of roles and responsibilities [98,99]. Individuals often overlook self-care while balancing work, family life, and recreation, leading to emotional distress and exhaustion [100]. Changes in leisure activities have been correlated with stress, depression, and overall well-being for family members [101,102]. Despite these challenges, family resilience has played a crucial role in supporting family functioning and promoting positive adaptation for all family members. It is a process that helps families overcome significant life challenges and return to their pre-crisis level of functioning [103,104,105]. Family resilience is rooted in a multi-level systems approach, acknowledging the impact of both internal family processes and external social contexts on family functioning [106]. Those with balanced levels of resilience within their families are better equipped to cope with the psychological effects of the pandemic, leading to better overall family functioning [107]. Thus, family resilience is an independent variable that drives family functioning and contributes to the overall functional performance of the family.

The purpose of this study is to examine family communication as a mediating factor. The Circumplex Model suggests that family communication is an important facilitator. Effective communication is essential for balancing resilience and cohesion and is the most sensitive indicator of family functioning [63,70]. Transparent communication is a protective factor, while non-transparent communication can exacerbate challenges [104]. Family communication is crucial in mediating the relationship between family resilience and family functioning [108]. Intra-family communication is critical to maintaining a healthy relationship between parents and children [90]. Personal communication between parents and children can also improve family functioning [109]. Communication, including expression and conflict resolution, affects family cohesion and adaptability [110]. Families can build resilience by establishing a family identity and history that promotes strength and adaptation, even in challenging experiences [111]. Communication is essential in achieving a balance of cohesion and flexibility in family relationships [112].

Previous studies have shown that communication plays a significant role in family resilience [111,112,113] and that there is a relationship between resilience and family functioning [114]. Communication can also mediate between many family variables [72,92,115,116]. The Core Model of this study indicates that families with high resilience will be better able to cope with the stress of quarantine during the pandemic in the Algeria and Iraq populations. The Circumplex Model suggests that family communication is critical to family functioning. As family communication improves, family resilience and functioning will likely be enhanced, mediating the direct relationship between family resilience and family functioning.

This study is of great importance as it explores crucial family variables such as family resilience, family functioning, and family communication, which act as mediating variables. This research is a novel contribution to the Arab context, which has seen a lack of studies encompassing these variables. Therefore, our study aims to bridge this research gap in the Arab milieu, especially during the rapidly changing political, economic, and social landscape caused by the COVID-19 pandemic. These changes significantly affect family processes, socialization, marital relationships, and stability in Algeria and Iraq [77,117].

This study aimed to examine the role of family communication (satisfaction towards positive communication among family members) as a mediator variable between family resilience and family functioning (coherence, interaction, commitment, and intra-family coping) among a sample from Algeria and Iraq. It tested four hypotheses: (a) family resilience is positively related to family functioning, (b) family resilience is positively related to family communication, (c) family communication is positively related to family functioning, and (d) family communication significantly mediates the relationship between family resilience and family functioning.

## 2. Materials and Methods

### 2.1. Study Design

The current study implemented a cross-sectional design and convenience sampling. The study was conducted during the first and second waves of the COVID-19 pandemic (From May 2020 to January 2021) in Algeria and Iraq, where schools, universities, shops, and transportation were wholly closed during this period (the first year of the pandemic). Therefore, the survey was difficult to conduct via an in-person approach, and the online questionnaire was used via Google Form. The online survey was distributed via social media sites (e.g., Facebook, Messenger, and WhatsApp) and mass emails.

### 2.2. Sample and Procedure

Participants for this research were recruited through an electronic form due to the pandemic, using convenience sampling methods. Participation in the research was completely voluntary. Only individuals residing in Algeria and Iraq who had experienced quarantine during the pandemic, were interested in the research, and were over 18 were included in the study. Participants were informed that their participation was voluntary and that their data and responses would only be used for scientific research. Informed consent was obtained from all the participants. The ethical approval for conducting this study was also obtained from the Ethical Research Committee, Faculty of Human and Social Sciences, Hassiba Benbouali University of Chlef, Algeria, with reference number I05L03UN020120200002/01/2021, dated 1 January 2021. This study was conducted following the Declaration of Helsinki and its subsequent amendments.

The sample consisted of 359 participants (226 from Algeria (63%) and 133 from Iraq (37%)), aged between 18 and 50 years. The percentage of males (n = 155, 43.2%) was less than that of females (n = 204, 56.8%). Ages were variable; the largest group was aged 31~40 years (n = 128, 35.7%), followed by 24~30 years (n = 116, 32.3%), 41~50 years (n = 80, 22.3%), and 18~23 years (n = 35, 9.7%). Concerning marital status, the largest percentages were in the single group (n = 173, 48.2%) and the married group (n = 175, 48.7%), while the divorced group had a small percentage (n = 11, 3.1%). With regards to the academic level of the sample, the largest percentage were at a doctorate level (n = 124, 34.5%), followed by university level (n = 118, 32.9%), Master’s level (n = 94, 26.2%), and finally secondary and intermediate levels, respectively (n = 10, 2.8%) and (n = 4, 1.1%) (Table 1).

### 2.3. Instruments

#### 2.3.1. Family Communication Scale

The family communication scale (FCS) of Olson et al. [118] was adopted. The FCS consists of 10 items, and each item was assessed using a 5-point Likert scale (1: not at all, 5: very well). Family communication was measured by items such as: “Family members are attentive listeners”, “Family members can engage in calm discussions about problems”, and “Family members openly express their true feelings to each other”. A high level indicates how family communication is effective within the family. Olson et al. (2004) [118] reported an acceptable level of internal consistency (α = 0.88). Moreover, the FCS was found to have good internal consistency in our pilot samples: α = 0.92 for an adult sample and 0.91 for a parent sample.

#### 2.3.2. Family Resilience Questionnaire (WFRQ)

Walsh [119] developed the family’s resilience scale, in which 32 items were identified on a 5-point Likert scale (1 = rare; 5 = usually) followed by an open question. These 32 items are divided into nine dimensions: Family Belief System (Making meaning of adversity, Positive outlook, Transcendence and spirituality), Family Organizational Processes (Flexibility, Connectedness, and Mobilize social and economic resources(, and Communication and Problem-solving Processes (Clear, consistent information, Open emotional sharing, and Collaborative problem solving). When measuring family resilience, certain items are used to evaluate different aspects. For example, the Family Belief System is assessed with questions like “We face our difficulties together, as a couple/family, rather than separately”; Family Organizational Processes are evaluated with items such as “We are flexible in adapting to new challenges”; and for measuring Problem-Solving processes, questions like “We collaborate in exploring possibilities and in making decisions, and we handle disagreements fairly”, are used. The latter appertains to asking patients and relatives to determine any other aspects that could help them to face the crisis. The questionnaire has been translated into several environments, as it has achieved high reliability in different cultures, including the Iranian version of the scale [120], the Italian [121], and the Chinese version [122]. The Arabic version of the scale (Algeria and Iraq), translated by Sabah et al. [77], confirmed the validity and factor structure of the Arabic WFRQ using confirmatory factor analysis. In the current study, the values of Cronbach’s alpha were good: 0.886, 0.825, and 0.911.

#### 2.3.3. Family Functioning Scale

The scale was prepared by Aiche Sabah [84]; this scale was built using the perspective of assessing family strengths in family functioning [123]; in this view, light is shed on the qualities of strong families and approaching strengths assessment from a proactive and promotional perspective, assessing and intervening in ways that build on existing competencies to strengthen the family unit and the individual family members’ functioning. Also, reviewing studies relied on some theoretical frameworks and measures that dealt with family functioning [63,124,125]. Nonetheless, after studying perceptions of the subject, the final version of the scale was determined, as it consists of four main dimensions: Coherence, Interaction, Commitment, and Intra-family coping strategies. Each dimension consists of four items to become 16 items. The scale statements are rated using a 5-point Likert scale (1 = never, 2 = rarely, 3 = sometimes, 4 = often, and 5 = always), and higher degrees indicate better family functioning. In the context of the family functioning scale, the following example items were used to measure various aspects of family functioning: Cohesion: “My family members seek help from each other”, Interaction: “Interactions among my family members are characterized by mutual trust and honesty”, Commitment: “My family members prioritize spending time together despite their busy schedules”, and Intra-family coping strategies: “We have multiple alternatives for resolving our problems within the family”. The scale’s psychometric properties were found to be satisfactory [84]. In this study, the values of Cronbach’s alpha were good: 0.784, 0.851, 0.811, and 0.745.

### 2.4. Statistical Analyses

IBM SPSS Statistics Version 28 was used for descriptive statistics, while AMOS program Version 24 was employed for structural equation modeling (SEM) to examine the relationships between the variables. The choice of utilizing SEM in this study is rooted in its ability to account for measurement errors when assessing associations simultaneously. In contrast, regression models, while capable of determining associations, do not incorporate measurement errors. Therefore, SEM, as a statistical technique, establishes measurement models and structural models to analyze complex behavioral relationships [11,26,126,127]. This study used the Structural Equation Model with the Maximum Likelihood method to explore the structural model linking family resilience as an independent variable, family functioning as a dependent variable, and family communication as a mediator variable. Additionally, bootstrapping in AMOS (number of bootstrap samples = 1000) was employed to test the mediation and confirm this hypothesis. The validity of the measurements in this study was assessed using goodness-of-fit measures, including the significance level of the Chi-Square statistic (non-significant), the comparative fit index (CFI) (≥0.90), the standardized root mean square residual (SRMR) (≤0.08), and the root mean square error of approximation (RMSEA) [11,17,126,128].

## 3. Results

### 3.1. Preliminary Variable Analysis

To ensure the normality of distribution, the skewness and kurtosis were calculated, and the values of the three variables were shown to be within an acceptable range (−2/+2). The correlation between the three variables was calculated, and all correlations were positively linked. Table 2 shows the preliminary variable analysis.

### 3.2. Structural Model

Figure 1 shows the proposed structural model for family resilience associated with family functioning via family communication as a mediator variable. The fit indices in the SEM indicated that the CFI was 0.954, the SRMR 0.041, and the RMSEA 0.057.

Table 3 displays the Amos outputs for all direct relationships: (a) there is a positive relationship between family resilience and family functioning (effect size of standardized coefficient = 0.760; *p* < 0.001), (b) there is a positive relationship between families’ resilience and family communication (effect size of standardized coefficient = 0.749; *p* < 0.001), and (c) there is a positive relationship between family communication and families’ functioning (effect size of standardized coefficient = 0.187; *p* = 0.001). The regressions showed that all the effects are significant at a level less than 0.05; therefore, the hypotheses are supported with statistical significance.

#### 3.2.1. Testing Mediation Using Bootstrapping in AMOS

By using bootstrapping in AMOS, the following results were found:AThe value of Total Effects was estimated at 0.808 and it is statistically significant (*p* = 0.002). Lower Bounds (BC) = 0.735. Upper Bounds (BC) = 0.886.BThe value of Direct Effects was 0.682 and it is statistically significant (*p* = 0.004). Lower Bounds (BC) = 0.558. Upper Bounds (BC) = 0.793.CThe value of Indirect Effects was 0.126 and it is statistically significant (*p* = 0.005). Lower Bounds (BC) = 0.053. Upper Bounds (BC) = 0.219.DMediation: The table showed that total, direct, and indirect effects are statistically significant. This means that the family communication variable mediates the relationship between families’ resilience and families’ functioning. On the other hand, the type of this mediation is partial because the direct relation has been shown to be statistically significant; family communication is a partial mediator variable between family resilience and families’ functioning.

#### 3.2.2. Multi-Group Analysis between Iraq and Algeria Samples

Although the results in Table 4 of multi-group SEM had significant χ^2^, other fit indices were acceptable or close to acceptable: CFI ranged between 0.899 and 0.906, and RMSEA between 0.058 and 0.060. Accordingly, the structural model comparison showed that the nested models were not significantly different (*p*-values between 0.687 and 0.750). Therefore, there are no differences between the sample models of Iraq and Algeria.

## 4. Discussion

The purpose of the current study was to investigate the relationship between family resilience, family communication, and family functioning. Also, the present study aimed to assess whether family communication was a mediator in the association between family resilience and family functioning. The study’s results indicated a positive relationship between family resilience, communication, and functioning. Regarding the role of family communication as a mediator, partial mediation of family communication in the association between family resilience and family functioning was observed.

In general, the findings from this study increase comprehension regarding the role of family resilience and communication in developing family functioning; likewise, the role of communication is like a facilitating indicator in the context of family [69,70,93,94,105]. The study’s results align with Olson’s Circular Model [69,70] which incorporates communication as a facilitator between flexibility and cohesion. This model has shown its effectiveness in various cultures. Additionally, the study’s findings are consistent with Rini’s research [108], highlighting the importance of effective communication among family members in promoting resilience and overall family performance. Moreover, the results have indicated no differences between the samples of Iraq and Algeria, which indicates that the Arab environment shares family process across countries. Resilience refers to the ability to bounce back from adversity. Family resilience may lead to positive adaptation and it may further refer to dealing with a complex situation of adversity [98]. Additionally, resilient individuals can effectively and ideally deal with adverse situations, such as the COVID-19 pandemic, without affecting family communication.

Regarding the role of family communication as a mediator, it appears this mediation could be partially but essential. Walsh [98] demonstrates that communication is a family process facilitating immediate and long-term adaptation to a family crisis. Above and beyond, family communication has a crucial role in the family’s functioning, as confirmed by Olson’s [93] Circumplex Model. Families that communicate to balance cohesion and resilience have more resilient individuals regardless of negative experiences. Furthermore, how families communicate in the face of stressful events likely influences people’s ability to cope with these events. Hence, through communication, families teach children how to control their emotions, manage stress, and cope with hardship [112].

Furthermore, family communication has a partial effect as an intermediate variable between family resilience and family functioning because of the vital role of family resilience in family functioning. Previous studies have agreed with a mediating role of family communication [72,92,115,116], for example, the theory of family resilience [98,119] that focuses on adaptation at a hard time, and Olson’s (2011) Circumplex Model that considers communication as a facilitating dimension helping families to develop cohesion and resilience to better deal with developmental demands. Communication is a fundamental process that makes it easier for families to cope with the crisis caused by the COVID-19 pandemic and the accompanying grief, trauma, and loss. According to Walsh [119] and Greeff and Human [129], openly and honestly communicating is essential to a family’s resilience throughout the loss process, especially in the face of transitional challenges in the immediate follow-up. Hence, open and honest communication is a necessary element of grief resolution.

With regards to the differences between Algeria and Iraq, the comparison indicators between the Iraq sample and the Algerian sample showed that there were no differences between them. Likewise, these results confirm that families in Algeria and Iraq face different crises through communication and resilience. They collectively face challenges by using family communication and interacting as a unit, not as separate individuals. This method of interaction and communication is applied in families through which they face crises. It is an extension of the coherence and dealing with crisis collectively within Arab families.

### 4.1. Implications

Concerning the theoretical implications of the study, this study could contribute to the literature on family psychology in the Arab environment during pandemics and crises. This study can develop the literature on family processes during the pandemic by expanding research on family communication and the role of facilitating other family processes, such as family resilience and functioning, in facing crises resulting from the outbreak of COVID-19. Moreover, this research fills an important gap in Arab research, which focused most research on studying each variable separately. In contrast, our study dealt with the relationships of essential family variables (communication, resilience, and family functioning) with each other by addressing the role of communication as a mediator. Similarly, our results give credibility to the role of family communication as a facilitating and mediating factor in crises and pandemics, thus lending credibility and further evidence of the ability of the Circumplex Model to interpret critical family processes.

In terms of practical implications, the study revealed a positive correlation between family resilience, family communication, and family functioning. This highlights why assessing these aspects during crises is important to help families navigate tough situations. The Circumplex Model, which identifies communication as a crucial factor in various family processes, is useful for understanding how families respond to crises. The findings emphasize the critical roles of family resilience and communication in enhancing overall family performance. Therefore, programs and interventions to strengthen family relationships and functioning should prioritize promoting resilience and effective communication within the family. The study also highlights the importance of communication as a facilitating factor in the family context. Initiatives to improve family dynamics should focus on strategies for effective communication, as it can foster cohesion and resilience within the family unit. Moreover, the absence of significant differences between the Iraqi and Algerian samples implies that family processes and dynamics are generally consistent across diverse Arab countries. This indicates the potential to develop programs that promote family functioning and well-being within various Arab cultural contexts.

### 4.2. Limitations of the Study

Despite the significance and alignment of this study with the theoretical framework, the present study has the following limitations. First, the nature of the pandemic differs from other epidemics that humans have experienced. In particular, the responses to this pandemic may be prolonged and result in modifying or altering various family processes. Consequently, the current findings are not conclusive and it is important to note that these results may change significantly after the pandemic concludes. Therefore, the generalization of these findings should be approached cautiously. Second, this study was conducted during the first wave of the COVID-19 pandemic. The responses to the pandemic and families’ strategies to cope with it may evolve during subsequent waves and even after the pandemic subsides. Third, the present study exclusively utilized the quantitative method, which limited the ability to explore findings that are typically attainable through qualitative research. Fourth, the study was cross-sectional, meaning the results could be influenced by a lack of temporal precedence among the variables. Therefore, there are limitations to the generalizability of the findings. We suggest that future studies use longitudinal or experimental research designs to address these limitations.

## 5. Conclusions

In conclusion, family communication could be the key to maintaining family functioning and the mental health of family members, which helps the family grow after the trauma of the COVID-19 pandemic. The results showed that family communication mediated the relationship between family resilience and family functioning, and there is a significant direct relation between family resilience and family functioning. According to the findings, healthcare providers should consider improving family resilience and communication to achieve good family functioning. Furthermore, future intervention studies should focus on family communication to maintain the psychological health of the family members during crises, pandemics, and other things.

## Figures and Tables

**Figure 1 children-10-01742-f001:**
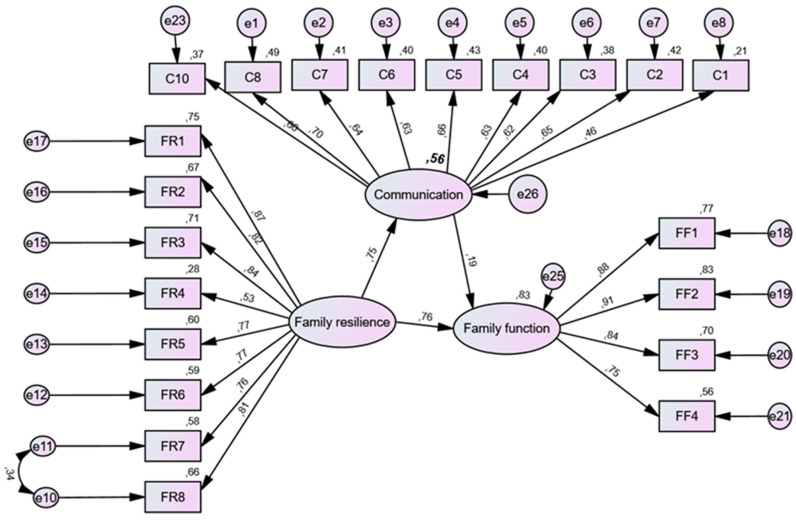
Structural model.

**Table 1 children-10-01742-t001:** Socio-demographic characteristics of the sample.

Item	n	%
Country		
Iraq	133	37.0
Algeria	226	63.0
Gender		
Male	155	43.2
Female	204	56.8
Age		
18–23 years	35	9.7
24–30 years	116	32.3
31–40 years	128	35.7
41–50 years	80	22.3
Marital status		
Single	173	48.2
Married	175	48.7
Divorced	11	3.1
Educational level		
Primary school	10	2.8
Middle school	4	1.1
Secondary school	9	2.5
University	118	32.9
Master’s degree	94	26.2
Doctorate degree	124	34.5

**Table 2 children-10-01742-t002:** Means, standard deviations, normality of distribution, and correlations among variables.

Variables	M	SD	Skewness	Kurtosis	r		
					Family Resilience	Family Function	Family Communication
Family resilience	122.95	19.43	−0.542	0.084	–		
Family function	59.91	10.83	−0.397	−0.088	0.830 **	–	
Family communication	36.82	5.86	−0.521	0.565	0.680 **	0.669 **	–

Notes. ** *p* < 0.01; possible score range for family resilience is between 32 and 160 (i.e., summing up 32 item scores); for family function is between 16 and 80 (i.e., summing up 16 item scores); and for family communication is between 10 and 50 (i.e., summing up 10 item scores).

**Table 3 children-10-01742-t003:** Associations between Family Communication, Family Resilience, and Family Functioning.

Variables			Regression Weights				Standardized Regression Weights
			Estimate	S.E.	t	*p*	Estimate
Communication	<---	Family Resilience	0.109	0.014	7.92	<0.001	0.749
Family Function	<---	Communication	1.153	0.358	3.22	0.001	0.187
Family Function	<---	Family Resilience	0.682	0.053	12.82	<0.001	0.760

**Table 4 children-10-01742-t004:** Model fit measures of multi-group analysis models.

	Model Fit Measures					
Models	χ^2^	DF	*p*	χ^2^/DF	CFI	RMSEA
M1. Unconstrained	773.683	336	<0.001	2.303	0.905	0.060
M2. Measurement weights	788.311	354	<0.001	2.227	0.906	0.059
M3. Structural covariances	788.977	355	<0.001	2.222	0.906	0.059
M4. Structural residuals	790.028	357	<0.001	2.213	0.906	0.058
M5. Measurement residuals	843.240	378	<0.001	2.231	0.899	0.059
**Model comparisons**	**Δχ^2^**	**ΔDF**	** *p* **		**ΔCFI**	**ΔRMSEA**
M1. vs. M2	14.628	18	0.69		0.001	−0.001
M2 vs. M3	0.666	1	0.41		0.000	0.000
M3 vs. M4	1.051	2	0.59		0.000	−0.001
M4 vs. M5	53.212	21	<0.001		−0.007	0.001

## Data Availability

The dataset supporting this study’s findings is not openly available and will be available from the corresponding author upon reasonable request.

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
