# Peer review of "Family Communication as a Mediator between Family Resilience and Family Functioning under the Quarantine and COVID-19 Pandemic in Arabic Countries"

_children, 2023, doi:10.3390/children10111742_

Round 1

Reviewer 1 Report

Comments and Suggestions for Authors

I thank the editors for providing the opportunity to review the manuscript entitled, “Family communication as a mediator between family resilience and family functioning under the quarantine and COVID-19 pandemic in Arabic countries”. I read the manuscript with great interest, however, there are significant concerns regarding the authors’ justification for the study, theorizing, analysis, interpretation of findings, and writing quality. While I was not able to provide a more positive review, I hope the recommendations below help the authors revise their manuscript for a more successful, future submission.

Study Justification and Contributions

Throughout the introduction and discussion, I was not able to discern a strong justification for the study. First, the first four paragraphs take up almost the whole first page but barely focus on the constructs under investigation. Second, I could not get a clear sense of what it is about family resilience, communication, and functioning that is worth studying with respect to the COVID-19 pandemic context and what the current study is contributing to the literature. Finally, the study was focused on adults aged 18-50, but the literature review was broader and non-specific to this age range. Perhaps this is an organizational issue, but the introductory pages would be more compelling if it was largely focused on the constructs (family resilience, communication, and functioning), processes (relations between family constructs), and the population (adults aged 18-50) under investigation. Doing so must also entail substantially shortening the rather long and meandering discussion on the impact of COVID-19 from the first three paragraphs. Finally, the gaps that the current study is filling must be clear from the beginning to get a sense of WHY this study is needed.

Theorizing

The authors must make a more compelling and clear argument regarding the processes under investigation. First, it is not clear either from the introduction or the method what the distinctions are between family functioning and family resilience, as their conceptualizations and operationalizations seem to be overlapping and/or muddled in the literature review and method. Second, and importantly, the authors need to make a clearer theoretical justification of the conceptual model, including convincing evidence why they conceive of family resilience as a predictor, communication as a mediator, and family functioning as an outcome. Section 1.3 does not make these distinctions clear at all and often makes arguments that contradict the model they’re analyzing (e.g., “the most sensitive indicator of family functioning and a crucial predictor of family resilience has been shown to be communication”.).

Analysis

I found aspects of the analysis a bit unclear and in need of rigor. First, sampling procedures need to be described in more detail than that in the current manuscript. Second, representative items from the instruments need to be listed to help readers get a sense of what the constructs under investigation actually are. After reading the introduction and Method, I don’t have a concrete idea of what the authors are measuring given in large part to the lack of sample items. Third, the authors don’t seem to give mention of important background or demographic covariates that they included on the analysis. Given the sociodemographic diversity of the sample in Table 1, variation in the results may be due to these exogenous factors, but it reads as if they were left out in the analysis. Fourth, it is not clear how the authors created the main variables. The scale of the items ranged from 1 to 5, yet their means are beyond this range (36.82 to 122.95, making it seem like these are summation scores). More detail is needed to understand how the authors came up with the composite variable scores. Fifth, Table 4 needs to include the changes in chi-squared, changes in degrees of freedom, and the p-values of these changes across the different models tested.

Interpretation of Findings

The authors may do well to avoid over-emphasizing the promise of their findings amid the theoretical and analytical limitations of the study. Beyond what were mentioned above, their study was cross-sectional in nature. As such, findings cannot speak unequivocally about effects, mediation, or “x predicting y” given the lack of temporal precedence in the variables, the discounting of key covariates, and the observational (versus experimental or quasi-experimental) data. However, these factors are neither even discussed in the Limitations section, nor did it hinder the authors from making broad-sweeping, overpromising claims in the Implications section. The authors need to take better care of the emphases in their claims in the face of clear analytic limitations.

Writing Quality

As a minor comment, I hope the authors revise the manuscript closely and watch for grammatical errors, flow issues, and opportunities for concision.

Comments on the Quality of English Language

As a minor comment, I hope the authors revise the manuscript closely and watch for grammatical errors, flow issues, and opportunities for concision.

Author Response

Reviewer 1

I thank the editors for providing the opportunity to review the manuscript entitled, “Family communication as a mediator between family resilience and family functioning under the quarantine and COVID-19 pandemic in Arabic countries”. I read the manuscript with great interest, however, there are significant concerns regarding the authors’ justification for the study, theorizing, analysis, interpretation of findings, and writing quality. While I was not able to provide a more positive review, I hope the recommendations below help the authors revise their manuscript for a more successful, future submission.

Response:

We wanted to express our gratitude to you for reviewing our manuscript. Your feedback, suggestions, and recommendations were thoughtful, and we sincerely appreciate them. We carefully addressed all the points you raised in our revised manuscript and provided a detailed response in this letter. Thank you again for your valuable input.

Point 1:

Study Justification and Contributions

Throughout the introduction and discussion, I was not able to discern a strong justification for the study.

First, the first four paragraphs take up almost the whole first page but barely focus on the constructs under investigation.

Response 1:

Thank you very much for the valuable comments. As the study focuses on "Family communication as a mediator between family resilience and family functioning under the quarantine and COVID-19 pandemic", we initially presented the topic within the context of the COVID-19 pandemic, we highly appreciate your comments to help us improve the introduction. Following your suggestions, we have now shortened the introduction and rewritten it concisely. We have also tried to make it clearer and more informative. Your feedback has been very helpful, and we thank you for your time and effort.

Point 2:

Second, I could not get a clear sense of what it is about family resilience, communication, and functioning that is worth studying with respect to the COVID-19 pandemic context and what the current study is contributing to the literature.

Response 2:

Thank you for your valuable feedback. We have now restructured the introduction based on your suggestions. The revised version now clearly defines family resilience, communication, and functioning and emphasizes how our study contributes to the existing literature. You can review the changes made to the introduction in the revised manuscript, marked with track changes.

Point 3:

Finally, the study was focused on adults aged 18-50, but the literature review was broader and non-specific to this age range.

Response 3:

Thank you for the valuable point. We have now revised the literature review to reflect the study's target age better.

Point 4:

Perhaps this is an organizational issue, but the introductory pages would be more compelling if it was largely focused on the constructs (family resilience, communication, and functioning), processes (relations between family constructs), and the population (adults aged 18-50) under investigation. Doing so must also entail substantially shortening the rather long and meandering discussion on the impact of COVID-19 from the first three paragraphs. Finally, the gaps that the current study is filling must be clear from the beginning to get a sense of WHY this study is needed.

Response 4:

Thank you for providing us guidance in the reorganization to present our research idea. Via your organizational insights, we have now reorganized the introduction in a more structured manner. Based on your suggestions, we have now made the following changes: we shortened the initial four paragraphs discussing COVID-19 into one to avoid redundancy. We presented the impact of the pandemic on families concerning the study's constructs - family resilience, communication, and functioning - and the processes (relationships between family constructs), investigating the study population (adults aged 18-50). At the end of the introduction, we clearly stated the study's gaps, rationale, objectives, and hypothesis for the current study. Please refer to the introduction to view the changes we made.

Point 5:

Theorizing

The authors must make a more compelling and clear argument regarding the processes under investigation. First, it is not clear either from the introduction or the method what the distinctions are between family functioning and family resilience, as their conceptualizations and operationalizations seem to be overlapping and/or muddled in the literature review and method.

Response 5:

Thank you for pointing out the theorizing issue. We have now made several revisions to this section, enhancing its clarity and coherence.

Firstly, we have now introduced Olson's Circumplex Model, which addresses the three variables. We then delved into family resilience by defining it and discussing previous studies conducted during the pandemic. Following that, we defined family functioning and explored its components in the study context. Finally, we discussed family communication by providing a definition, referencing previous studies, and elaborating on its role as a mediating variable.

Kindly see section 1.2 for the changes made based on your comments. Once again, we appreciate your valuable feedback, and we believe that it has helped us to improve the quality of our work..

Point 6:

Second, and importantly, the authors need to make a clearer theoretical justification of the conceptual model, including convincing evidence why they conceive of family resilience as a predictor, communication as a mediator, and family functioning as an outcome. Section 1.3 does not make these distinctions clear at all and often makes arguments that contradict the model they’re analyzing (e.g., “the most sensitive indicator of family functioning and a crucial predictor of family resilience has been shown to be communication”.).

Response 6:

We have now carefully revised and reworded Section 1.3 based on your comments. This section is supported by studies that endorse the study model, which considers resilience as an independent variable, communication as a mediator, and family functioning as a dependent variable. This division is derived from the theoretical model proposed by Olson, which emphasizes communication as a key facilitator. Please refer to Section 1.3 to review our changes based on your feedback.

Point 7:

I found aspects of the analysis a bit unclear and in need of rigor. First, sampling procedures need to be described in more detail than that in the current manuscript.

Response 7:

We have now added further details to the sampling procedures section of the research paper. Due to the pandemic, participants were recruited through an electronic form using convenience sampling. The inclusion criteria for the study required participants to be over 18 years of age, reside in Algeria or Iraq, have experienced quarantine during the pandemic, and express interest in the research. We informed participants that their participation was voluntary and that their data and responses would be used solely for scientific research. We also obtained informed consent from all participants.

Point 8:

Second, representative items from the instruments need to be listed to help readers get a sense of what the constructs under investigation actually are. After reading the introduction and Method, I don’t have a concrete idea of what the authors are measuring given in large part to the lack of sample items.

Response 8:

We have now added representative items in the instruments section to aid readers' understanding of the study variables and instruments.

Point 9:

third, the authors don’t seem to give mention of important background or demographic covariates that they included on the analysis. Given the sociodemographic diversity of the sample in Table 1, variation in the results may be due to these exogenous factors, but it reads as if they were left out in the analysis

Response 9:

Thank you for your valuable observation. While we appreciate that the study includes a wealth of demographic variables that could potentially enhance the research, the primary goal of our study was to calculate mediation through the proposed model and investigate differences in the proposed model within the contexts of Algeria and Iraq.

Point 10:

Fourth, it is not clear how the authors created the main variables. The scale of the items ranged from 1 to 5, yet their means are beyond this range (36.82 to 122.95, making it seem like these are summation scores). More detail is needed to understand how the authors came up with the composite variable scores.

Response 10:

Thank you for your comment. To provide a better understanding of the data, I would like to clarify that the family communication scale ranges from 10 to 50, the family resilience scale spans from 32 to 160, and the family functioning scale ranges from 16 to 80. After collecting the data and calculating their arithmetic averages, the scores were from 36.82 to 122.95. Such information is now provided in Table 1 footnotes.

Point 11:

Fifth, Table 4 needs to include the changes in chi-squared, changes in degrees of freedom, and the p-values of these changes across the different models tested.

Response 11:

Thank you very much for your comment. All the suggested results now are included in Table 4.

Point 12:

The authors may do well to avoid over-emphasizing the promise of their findings amid the theoretical and analytical limitations of the study. Beyond what were mentioned above, their study was cross-sectional in nature. As such, findings cannot speak unequivocally about effects, mediation, or “x predicting y” given the lack of temporal precedence in the variables, the discounting of key covariates, and the observational (versus experimental or quasi-experimental) data. However, these factors are neither even discussed in the Limitations section, nor did it hinder the authors from making broad-sweeping, overpromising claims in the Implications section. The authors need to take better care of the emphases in their claims in the face of clear analytic limitations.

Response 12:

I appreciate your valuable observation. The study was conducted during the COVID-19 pandemic, which posed specific research conditions related to applying studies. During that time, online applications and cross-sectional studies became more prevalent, while experimental and quasi-experimental studies were challenging. Nonetheless, this does not invalidate the importance and validity of cross-sectional studies. Therefore, we have now included this information in the limitations section of the study.

Point 13:

Writing Quality

As a minor comment, I hope the authors revise the manuscript closely and watch for grammatical errors, flow issues, and opportunities for concision.

Comments on the Quality of English Language

As a minor comment, I hope the authors revise the manuscript closely and watch for grammatical errors, flow issues, and opportunities for concision.

Response 13:

Thank you for the helpful comments that improved and organized the manuscript. The language was proofread.

Reviewer 2 Report

Comments and Suggestions for Authors

The study is correct but is still subject to a number of modifications to be made:

- The introduction involves sufficient studies on covid but forgets to provide information regarding family variables and their relevance. In addition, the state of the art, the review of previous studies is very scarce making it difficult to make visible the knowledge gap that motivates the present study.

- The type of sampling is not described, if it has been carried out under the sample calculation formula to guarantee the generalisability of the results. Furthermore, there is no mention of the inclusion and exclusion criteria.

- A procedural section describing the whole process and ethical issues with the sample is missing. 

- In the discussion there are parts that are not discussed, i.e. there is an absence of studies to make a comparison between what was previously done and what is represented in these results. 

- The results do not show the effect size, which describes the magnitude of the associations found.

- The references are missing some references on family leisure in covid or previous references on family functioning.

Author Response

Reviewer 2:

The study is correct but is still subject to a number of modifications to be made:

Response:

Thank you very much for your incredible efforts and positive feedback. We deeply appreciate your thorough review and comments to our manuscript. By using your insightful and valuable comments, we have improved the quality of our manuscript. In this letter, our point-by-point responses will be presented below.

Point 1:

The introduction involves sufficient studies on covid but forgets to provide information regarding family variables and their relevance. In addition, the state of the art, the review of previous studies is very scarce making it difficult to make visible the knowledge gap that motivates the present study.

Response 1:

Thank you for your valuable feedback. We have now thoroughly revised the introduction and the theoretical framework to address your comments. Specifically, we have now highlighted the three variables in the introduction, focusing on the research sample. Also, we have now provided clear definitions of the variables in the second section, along with a review of previous studies on each variable during the pandemic. Furthermore, we have now clarified how these variables fit within the Olson Circumplex Model. At the end of the introduction, we clearly stated the study's gaps, rationale, objectives, and hypothesis for the current study.

Point 2:

The type of sampling is not described, if it has been carried out under the sample calculation formula to guarantee the generalisability of the results. Furthermore, there is no mention of the inclusion and exclusion criteria.

Response 2:

Thank you very much for your valuable feedback. We have now incorporated additional details into the sample section of our research. However, we acknowledge that the generalizability of our findings may be limited due to the convenience sampling method we used. Thus, this is listed as a limitation in our limitations.

Here is the corrected sample and procedure section:

“Participants for this research were recruited through an electronic form due to the pandemic, using convenience sampling methods. The participation in the research was completely voluntary. Only individuals residing in Algeria and Iraq who had experienced quarantine during the pandemic were interested in the research and were over 18 were included in the study. Participants were informed that their participation was voluntary and that their data and responses would only be used for scientific research. The informed consent was obtained from all the participants. The ethical approval for conducting this study has been obtained also from the ethical research Committee, Faculty of Human and Social Sciences, Hassiba Benbouali University of Chlef, Algeria, with reference number I05L03UN020120200002/01/2021, dated 1 January 2021. This study was conducted following the Declaration of Helsinki and its subsequent amendments.”

Point 3:

A procedural section describing the whole process and ethical issues with the sample is missing.

Response 3:

Thank you very much for your comment. We have now added the procedural details into the sample and procedure section.

Here is the corrected sample and procedure section:

“Participants for this research were recruited through an electronic form due to the pandemic, using convenience sampling methods. The participation in the research was completely voluntary. Only individuals residing in Algeria and Iraq who had experienced quarantine during the pandemic were interested in the research and were over 18 were included in the study. Participants were informed that their participation was voluntary and that their data and responses would only be used for scientific research. The informed consent was obtained from all the participants. The ethical approval for conducting this study has also been obtained from the ethical research Committee, Faculty of Human and Social Sciences, Hassiba Benbouali University of Chlef, Algeria, with reference number I05L03UN020120200002/01/2021, dated 1 January 2021. This study was conducted following the Declaration of Helsinki and its subsequent amendments.”

Point 4:

-In the discussion there are parts that are not discussed, i.e. there is an absence of studies to make a comparison between what was previously done and what is represented in these results. Response 4:

Thank you for providing the directions to elaborate our Discussion section. Comparisons have been added to compare previous work to the presented results. For example, as follows:

“The study results align with Olson's Circular Model [69,70] that incorporates communication as a facilitator between flexibility and cohesion. This model has shown its effectiveness in various cultures. Additionally, the study's findings are consistent with Rini's research [108], highlighting the importance of effective communication among family members in promoting resilience and overall family performance.”

Point 5:

The results do not show the effect size, which describes the magnitude of the associations found.

Response 5:

Thank you for pointing this out. In structural equation modelling, the standardized coefficients are considered the path's effect size. We have now clearly mentioned this in the Results section and Table 3.

“Table 3 displays the Amos outputs for all direct relationships (a) there is a positive relationship between family resilience and family functioning (effect size of standardized coefficient = 0.760; p < 0.001), (b) there is a positive relationship between families’ resilience and family communication (effect size of standardized coefficient = 0.749; p < 0.001), (c) there is a positive relationship between family communication and families functioning (effect size of standardized coefficient = 0.187; p = 0.001).”

Point 6:

The references are missing some references on family leisure in covid or previous references on family functioning

Response 5

Thank you for the valuable feedback. We have now added some studies on family leisure in COVID-19 and on family functioning.

Reviewer 3 Report

Comments and Suggestions for Authors

This research explores the role of family communication as a mediator between two variables, i.e., family resilience and family functioning during the COVID-19 pandemic in Arabic countries. The topic is engaging and the investigation of the family psychological dynamics in such a distinct epidemiologic context may interest the journal’s readers. A thorough proofreading is needed, because some syntactical aspects need to be addressed for a better readability of the text. Also, please refer to the following observations:

Line 24- please put „FFS” between brackets, for consistency;

Line 84- „researches has...” is grammatically incorrect;

Lines 85-86- please rephrase this segment: „...with long-term effects, which could have long-lasting effects”;

Line 101- the subject is missing: the effect of what? Also, where is the sub-chapter 1.1;

Line 110, 121- after a colon, it is not customary to proceed with a capital letter;

Line 157- Sabah needs a capital letter;

Line 165-166- the significance of the sentence ”the family can charge strength when 165 their life is at risk” is not very clear;

Line 225- please use the plural form for ”Instrument”;

Line 234- please use capital letters for “Family” and “Resilience”;

Lines 328-331- are redundant, and maybe they remained from the initial template, please consider eliminating them from the manuscript.

Formulating some more specific practical consequences of this research’s results would be beneficial for increasing the readers' interest.

Comments on the Quality of English Language

Proofreading is needed

Author Response

Reviewer 3:

This research explores the role of family communication as a mediator between two variables, i.e., family resilience and family functioning during the COVID-19 pandemic in Arabic countries. The topic is engaging and the investigation of the family psychological dynamics in such a distinct epidemiologic context may interest the journal’s readers.

Response:

Thank you so much for your remarkable efforts and positive feedback. We greatly value your comprehensive review and comments on our manuscript. Your insightful and valuable comments have significantly enhanced the quality of our paper. In this response letter, we provide detailed responses to your points.

Point 1:

The introduction involves sufficient studies on covid but forgets to provide information regarding family variables and their relevance. In addition, the state of the art, the review of previous studies is very scarce making it difficult to make visible the knowledge gap that motivates the present study.

Response 1:

Thank you for your valuable feedback. We have now thoroughly revised the introduction and the theoretical framework to address your comments. Specifically, we have now highlighted the three variables in the introduction, focusing on the research sample. Also, we have now provided clear definitions of the variables in the second section, along with a review of previous studies on each variable during the pandemic. Furthermore, we have now clarified how these variables fit within the Olson Circumplex Model. At the end of the introduction, we clearly stated the study's gaps, rationale, objectives, and hypothesis for the current study.

Point 2:

A thorough proofreading is needed, because some syntactical aspects need to be addressed for a better readability of the text. Also, please refer to the following observations:

Line 24- please put „FFS” between brackets, for consistency;

Line 84- „researches has...” is grammatically incorrect;

Lines 85-86- please rephrase this segment: „...with long-term effects, which could have long-lasting effects”;

Line 101- the subject is missing: the effect of what? Also, where is the sub-chapter 1.1;

Line 110, 121- after a colon, it is not customary to proceed with a capital letter;

Line 157- Sabah needs a capital letter;

Line 165-166- the significance of the sentence ”the family can charge strength when 165 their life is at risk” is not very clear;

Line 225- please use the plural form for ”Instrument”;

Line 234- please use capital letters for “Family” and “Resilience”;

Lines 328-331- are redundant, and maybe they remained from the initial template, please consider eliminating them from the manuscript.

Response 2:

Thank you for your valuable comments and for pointing out the language errors. After receiving your valuable feedback and noting some language errors, we have now made the necessary corrections and proofread the manuscript.

Point 3:

Formulating some more specific practical consequences of this research’s results would be beneficial for increasing the readers' interest.

Response 3:

Thank you for your feedback. We have now added the practical implications to the implications section.

“In terms of practical implications, the study revealed a positive correlation between family resilience, family communication, and family functioning. This highlights why assessing these aspects during crises is important to help families navigate tough situations. The Circumplex Model, which identifies communication as a crucial factor in various family processes, is useful for understanding how families respond to crises. The findings emphasize the critical roles of family resilience and communication in enhancing overall family performance. Therefore, programs and interventions to strengthen family relationships and functioning should prioritize promoting resilience and effective communication within the family. The study also highlights the importance of communication as a facilitating factor in the family context. Initiatives to improve family dynamics should focus on strategies for effective communication, as it can foster cohesion and resilience within the family unit. Moreover, the absence of significant differences between the Iraqi and Algerian samples implies that family processes and dynamics are generally consistent across diverse Arab countries. This indicates the potential to develop programs that promote family functioning and well-being within various Arab cultural contexts.”

Round 2

Reviewer 2 Report

Comments and Suggestions for Authors

There is not comments after the second version. Good job!